# The Impact of Gastrointestinal Symptoms on Patients’ Well-Being: Best–Worst Scaling (BWS) to Prioritize Symptoms of the Gastrointestinal Symptom Score (GIS)

**DOI:** 10.3390/ijerph182111715

**Published:** 2021-11-08

**Authors:** Axel Christian Mühlbacher, Anika Kaczynski

**Affiliations:** Institute for Health Economics and Health Care Management, Hochschule Neubrandenburg, 17033 Neubrandenburg, Germany; kaczynski@hs-nb.de

**Keywords:** with functional dyspepsia, best–worst scaling, patient preferences, preference-based score

## Abstract

Background: The gastrointestinal symptom score (GIS) is used in a standardized form to ascertain dyspeptic symptoms in patients with functional dyspepsia in clinical practice. As a criterion for evaluating the effectiveness of a treatment, the change in the summed total point value is used. The total score ranges from 0 to 40 points, in which a higher score represents a more serious manifestation of the disease. Each symptom is included with equal importance in the overall evaluation. The objective of this study was to test this assumption from a patients’ perspective. Our aim was to measure the priorities of patients for the ten gastrointestinal symptoms by using best–worst scaling. Method: A best–worst scaling (BWS) object scaling (Case 1) was applied. Therefore, the symptoms of the GIS were included in a questionnaire using a fractional factorial design (BIBD—balanced incomplete block design). In each choice set, the patients selected the component that had the most and the least impact on their well-being. The BIB design generated a total of 15 choice sets, which each included four attributes. Results: In this study, 1096 affected patients were asked for their priorities regarding a treatment of functional dyspepsia and motility disorder. Based on the data analysis, the symptoms abdominal cramps (SQRT (B/W): −1.27), vomiting (SQRT (B/W): −1.07) and epigastric pain (SQRT (B/W): −0.76) were most important and thus have the greatest influence on the well-being of patients with functional dyspepsia and motility disorders. In the middle range are the symptoms nausea (SQRT (B/W): −0.69), acid reflux/indigestion (SQRT (B/W): −0.29), sickness (SQRT (B/W): −0.26) and retrosternal discomfort (SQRT (B/W): 0.26), whereas the symptoms causing the least impact are the feeling of fullness (SQRT (B/W): 0.80), early satiety (SQRT (B/W): 1.54) and loss of appetite (SQRT(B/W): 1.95). Discussion: Unlike the underlying assumption of the GIS, the BWS indicated that patients did not weight the 10 symptoms equally. The results of the survey show that the three symptoms of vomiting, abdominal cramps and epigastric pain are weighted considerably higher than symptoms such as early satiety, loss of appetite and the feeling of fullness. The evaluation of the BWS data has illustrated, however, that the restrictive assumption of GIS does not reflect the reality of dyspeptic patients. Conclusions: In conclusion, a preference-based GIS is necessary to make valid information about the real burden of illness and to improve the burden of symptoms in the indication of gastrointestinal conditions. The findings of the BWS demonstrate that the common GIS is not applicable to represent the real burden of disease. The results suggest the potential modification of the established GIS by future research using a stated preference study.

## 1. Background

Gastrointestinal problems are widespread. The global prevalence of gastrointestinal disease is about 11.2% [1]. Clinically, gastrointestinal symptoms are divided into motility disorders and functional disorders of the gastrointestinal tract. Motility disorders of the gastrointestinal tract are very often associated with acute and chronic diseases of the gastrointestinal tract. Medical data indicate that worldwide, 30–45% of all GI conditions are referable to intestinal motility disorders. Motility disorders (“gastrointestinal problems”) are dysfunctions in the movement of the gastrointestinal tract, resulting in symptoms. Functional dyspepsia is a disorder of gastric function, predominantly leading to symptoms in the upper abdomen. For these, an organic cause cannot be found [2,3]. Both disorders are of social importance since they influence the well-being and quality of life of those affected [4,5,6,7,8]. Reliable methods of assessing symptom status are important for patient management as well as for treatment decisions since functional dyspepsia and motility disorder significantly disrupt patients’ lives [8]. Therefore, reliable methods of assessing symptom status are important for patient management as well as for treatment decisions [9].

The gastrointestinal symptom score (GIS) is an evaluation tool of gastrointestinal symptoms and is used in a standardized form to ascertain dyspeptic symptoms in patients with functional dyspepsia [10,11]. The disease-specific measurement has ten typical items for assessing functional dyspepsia gastrointestinal symptoms [11]. These symptoms are assessed individually based on a 5-point Likert scale from 0 to 4 by means of a face-to-face interview of the patient: 0—no complaints; 1—mild discomfort; 2—moderate discomfort; 3—severe discomfort; 4—very severe symptoms [11,12,13]. The point value of GIS consists of the sum of individual scores, which are assigned to the individual symptoms [10]. The total score (summed score) ranges from 0 to 40 points, a high score representing a serious manifestation of the gastrointestinal disease. As a criterion for evaluating the effectiveness of a treatment, the change in the total point value over the period of treatment is used. Therefore, the GIS is determined at the start of treatment on day 0 and after a certain number of weeks of treatment [11]. 

The GIS profile consequently serves to quantify the severity of symptoms, which can be used both in clinical studies and for the standardized measurement of symptoms in a clinical practice [11]. Accordingly, the usual GIS assumes that all ten gastrointestinal symptoms are weighted equally. Each symptom is included with equal importance in the overall evaluation.

Following the valuation of the GIS from Adam et al. in 2005, this instrument meets criteria of reproducibility, sensitivity, responsiveness and specificity for functional dyspepsia. Consequently, the authors concluded that the GIS profile is a reliable tool for assessing the symptoms of functional dyspepsia and the effectiveness of treatments. Furthermore, it allows an accurate assessment of the improvements in single symptoms [11]. However, it is to be noted that the GIS, which determines the burden of illness in patients with functional dyspepsia and motility disorder, is not preference-based. 

Within the GIS, weighting of the individual symptoms is required. The aggregation of the different symptoms needs information on how strong the symptoms are that should be included in the assessment and evaluation of the burden of disease. The measurement of various symptoms is only useful if a weighting of each single outcome is performed.

## 2. Objectives

The objective of this study was to test the underlying assumption that all ten gastrointestinal symptoms of the GIS are weighted equally from a patients’ perspective. The GIS does not consider any weighting of the included symptoms. If patients value symptoms differently, the score of the GIS does not truly reflect the burden of disease necessary to evaluate treatments related to gastrointestinal symptoms. Therefore, our aim was to analyze patients’ priorities and analyze differences in the impact that the ten gastrointestinal symptoms had on patients’ well-being. The meaning and importance of several symptoms from the perspective of the affected patients were tested and analyzed with a stated preference approach.

## 3. Methods

### 3.1. Study Design

The study was divided into two study phases: the qualitative pre-investigation determined the perspective of a gastrointestinal expert regarding the assessment of gastrointestinal symptoms [14,15]. In the principal investigation phase, the symptoms of the GIS were used as a basis for measuring and determining the relevance of each single outcome from a patient perspective.

#### 3.1.1. Pre-Investigation 

Prior to the main study, we performed an expert workshop to collect quantitative data from the experts that attended. The data collection was conducted in 2015 using a paper–pencil questionnaire among participants recruited in conjunction with the gastroenterological expert conference. Twenty experts from the field of gastroenterology took part in the survey [14,15].

#### 3.1.2. Principal Investigation

For further development of the GIS and for the valuation of the gastrointestinal measurement instrument, a best–worst scaling (BWS) method was tested and used in a patient survey. The main study was performed as an anonymous survey, which started in May 2015, using online questionnaires. Patients were contacted and sent replies to one of three designed online questionnaires. In total, 1096 patients participated in the survey. No personal data such as addresses, names or phone numbers were collected.

### 3.2. Method: Best–Worst Scaling (BWS)

As a form of discrete choice experiment (DCE) [16], best–worst scaling (BWS) is based on the assumption that respondents are able to judge the best and the worst (or the most important and the least important) out of three or more attributes or alternatives in a choice set [17,18]. The stated preference method is a multinominal expansion of the paired comparison method that has its basics in the random utility theory that dates back to the work of Thurstone (1927) [19]. In a traditional DCE, only the best alternative is selected. This methodological trait is avoided in BWS as it also collects information on the least preferred attribute or alternative [20,21,22].

BWS distinguishes three basic cases: object scaling (case 1), attribute or profile scaling (case 2) and multi-profiling (case 3) [16]. All three cases share the common assumption that respondents can choose the best and the worst or the most and least important from a set of at least three characteristics or alternatives. In the present study, object scaling (case 1) was used. For this, the individual attributes were first described and put together systematically in various choice sets [23]. This means that all choice tasks consisted of the same number of characteristics but varied in their combinations. In each choice set, the respondents chose the item with the strongest negative and the lowest negative impact on their well-being [20,21]. The design and the number of attributes determined the number of choice sets shown [22].

### 3.3. Decision Model

All attributes surveyed reflected the gastrointestinal symptoms of the common GIS [11,16]. The final decision model (attributes and their explanations) that was included in the final questionnaire is displayed in Table 1.

### 3.4. Data Collection and Recruitment

The survey focused on patients with functional dyspepsia (stomach irritation) and gastrointestinal motility disorders (gastrointestinal problems). The eligibility of patients was subject to certain inclusion and exclusion criteria. Accordingly, only the patients who met all the following criteria were eligible to participate in the survey:

Age: ≥18 years.

Diagnosis with functional dyspepsia or gastrointestinal motility disorders.

Outpatient or inpatient treatment due to functional dyspepsia or motility disorder in the last 6 months.

Informed consent.

By using a screening questionnaire, patients who did not meet all the inclusion criteria were ruled ineligible from the survey (“disqualified”). All participants who did not complete the questionnaire were also removed from the sample (“incomplete”).

The recruitment of patients was carried out by a third-party recruiting agency. The data collection was carried out in the period from June to November 2015. Data were collected through an online survey. For the created online questionnaires, Computer Assisted Personal Interviews (CAPIs) with the support of trained interviewers were completed by the patients.

### 3.5. Ethical Requirements and Approval

Regarding the legal and ethical requirements, all documents used in the study were reviewed through the Ethics Committee of the University of Greifswald by an application on 4 December 2014. By the decision on 28 January 2015, the trial was declared safe and released (BB 127/14).

It was an anonymous data collection. All subjects agreed to participate in the research project. All subjects were fully informed and aware of the research project. At any time of the survey, the participants were free to answer the questions addressed to them or to finish the questionnaire. The completion of the questionnaire was possible at any time.

### 3.6. Experimental Design 

A fractional factorial design was used to distribute the individual stimuli over the choice sets [24]. The BWS used a balanced incomplete block design in accordance with Cochran/Cox (1992) [25,26]. This design generated a total of ten choice sets, each including four attributes. The design was balanced, as each attribute appeared exactly the same number of times across all choice tasks. Each attribute was available for selection four times. Hence, a complete balance could be achieved. The allocation of respondents to the various questionnaire versions was randomized.

### 3.7. Survey Instrument

The questionnaire encompassed different domains:Sociodemographic characteristics (e.g., age, gender, educational level, previous therapy and several questions concerning health status, disease and treatment experiences).Explanation/description of the ten gastrointestinal symptoms.Questions regarding the frequency of occurrence of gastrointestinal symptoms using a 5-level Likert scale.Assessment of the impact of gastrointestinal symptoms on well-being with BWS.

For the survey, three different question versions were created. The questions presented when answering the BWS choice tasks varied between these three survey instruments. In the questionnaires, changing information sets were integrated. Thus, depending on the questionnaire version, the participants randomly received different information about the type of disease. The first questionnaire version presented acute disease complaints, the second version chronic disease complaints and the third questionnaire presented no information about the kind of complaints (reference). The aim of this approach was to control the perceptions via differentiation in the experiment. In addition, for each presented symptom, a visualization was included in the choice sets to provide an easier understanding of the characteristics. The randomization was not linked to any personal information. This was used in reference to the study hypothesis. In the GIS, all ten gastrointestinal symptoms were weighted equally, independent of the clinical status of the patients. Using this survey approach, this assumption should be tested using differing information sets, randomly assigned to patients with symptoms of varying severity.

### 3.8. Statistical Analysis

Sociodemographic data were analyzed using descriptive analyses such as frequency counts and statistical parameters of distributions and mean comparisons. The analysis of BWS data was carried out using count analyses. To achieve comparability of the results, the average best–worst score and the square root of the quotient of best and worst selections were calculated as well. SPSS, Sawtooth and Microsoft Excel were used. 

The results of the BWS needed to be re-coded in this study because of the research questions and the subject matter of the survey (GI symptoms). Hence, the symptom with the highest value (sqrt (b/w)) was not placed first in the ranking, but the symptom with the lowest value was. This is because in BWS, the symptom with the highest value has the least negative impact on patients’ well-being. The symptom with the lowest value has the greatest negative impact on patients. 

Based on the number of times a symptom was selected as best (i.e., the symptom with the least negative impact on patient well-being) or worst (i.e., the symptom with the most negative impact on patient well-being), the best–worst score was calculated [16].

Quantitative data analysis of the BWS included various procedures such as count analysis. [21] Best–worst scores were calculated showing the maximum difference between individual scores. The best–worst score was calculated by the subtraction of the two frequency results (best–worst = best–worst score) [22,27]. This B–W score can be used for statements on the importance and ranking of individual levels. To ensure complete results and better comparability, the B–W scores must be standardized. For this, the sample size (number of participants) was multiplied by the number of times a symptom was shown in the choice sets of the BWS (r = 4). The resulting product was then divided by the calculated best–worst score. This procedure is known as the average best–worst score. It allows for comparisons between different BWS data regardless of design and sample size [27]. Finally, the attributes were put in relation to one another based on the sum of all best and worst data. For this, the number of times the attribute was chosen as ‘best’ was divided by the number it was nominated ‘worst’. The resulting quotient was then square rooted (Sqrt) [16]. In a final step and for an easier, more appropriate interpretation of the results, the BW scores and the Sqrt (B/W) values were rescaled.

## 4. Results

### 4.1. Study Population

In total, 1096 patients with functional dyspepsia or motility disorders completed the BWS survey. The three versions of the questionnaire were spread evenly across respondents. Due to the randomized allocation of questionnaire version, 364 subjects completed the first questionnaire version (acute complaints), 358 subjects the second questionnaire version (chronic symptoms) and 374 subjects the third questionnaire version (no information).

In the sample, 45.6% (*N* = 500) of the participants were female. The age of participants ranged from 18 to 83 years. A total of 43.0% of respondents classified their current state of health as excellent or good. In total, 18.7% rated their health as less good or bad. 

Regarding the level of education, 51.2% claimed to have GCSEs or a college entrance. In total, 12.7% stated that they had completed a (technical) college or university degree. Most respondents (74.6%) were in a full- or part-time employment. 

The detailed socio-demographic characteristics are presented in Table 2.

In addition, an examination of the frequency of occurrence of the 10 gastrointestinal symptoms was used to validate the results of the BWS and to provide additional insight into the burden of illness of the patients. The analysis of the frequency of occurrence of the 10 gastrointestinal symptoms yielded the following results presented in Table 3.

In more than a third of the subjects (38.8%), the symptom “nausea” occurs only rarely (i.e., 1 time per month or less); however, in another third of respondents (32.3%), this gastrointestinal symptom occurs occasionally (i.e., more than 1 time per month). In total, 527 respondents and thus almost half of the participants (48.1%) indicated that the symptom “vomiting” never occurs. 

“Feeling of fullness”, “stomach cramps”, “early satiety”, “acid reflux/indigestion” and “upper abdominal pain” are the symptoms that occur most frequently in most patients surveyed.

### 4.2. Patient Priority Data

The evaluation of the gastrointestinal symptom score yielded the following results (Table 4).

For the patients surveyed, the symptoms of upper abdominal pain (Sqrt (B/W): −0.76), vomiting (Sqrt (B/W): −1.07) and abdominal cramps (Sqrt (B/W): −1.27) are most important and have greatest negative influence on patients’ well-being. Symptoms causing the least impact are the loss of appetite (Sqrt (B/W): 1.95), early satiety (Sqrt (B/W): 1.54) and the feeling of fullness (Sqrt (B/W): 0.80).

The following illustration (Figure 1) shows the rescaled best–worst scores of the gastrointestinal symptoms.

The level of the values reflects the impact of the symptoms on patients’ decisions. 

Regarding the different questionnaire versions, the analysis and comparison of the BWS results (i.e., questionnaire version with the information “acute complaints”, “chronic complaints” or “no information-reference”) shows almost a congruent priority order. For all three groups of patients, regardless of the kind of sickness (i.e., if suffering from acute or chronic complaints), the symptoms “vomiting” and “stomach cramps” are most important. These gastrointestinal symptoms show the highest negative impact on well-being. Both symptoms are dominant, displaying the strongest weights. The symptom “loss of appetite” is least important in all three groups. This gastrointestinal symptom has the least negative impact on well-being from the viewpoint of dyspeptic patients. In comparison, the other gastrointestinal symptoms have a similar weight in all three samples and show congruent results regarding the ranking of symptoms. The following figure (Figure 2) shows the weights (rescaled sqrt (best/worst) of the symptoms for all three study samples.

The results of the preliminary study regarding the importance of gastrointestinal symptoms from an expert’s perspective yields the following results in comparison to the patient survey [14]. On the one hand, it becomes clear that the individual symptoms of the GIS do not have the same meaning or importance in the experts’ point of view. The results show that even here, single outcomes are weighted differently by experts. On the other hand, the experts’ assessment regarding the influence of different symptoms on the well-being of the patients shows a slightly different ranking order than the patient’s assessment (see Figure A1 in Appendix A). For both groups, the symptoms “loss of appetite” and “early satiety” are ranked as the symptoms with the least negative impact on well-being. However, for the patients, the symptom “early satiety” is more important (i.e., shows a higher weight) than for the experts. In comparison, the experts weighted “loss of appetite” higher. In contrast, the symptoms “vomiting”, “stomach cramps” and “upper abdominal pain” are the symptoms with the most negative impact on patient well-being for both groups. From the experts’ view as well as from the patients’ view, these two symptoms have the least negative impact on well-being. Both symptoms show the highest weights. However, the symptom “stomach cramps” is ranked first in the patient survey, while in the experts’ survey, it is only ranked second. One of the biggest deviations between the assessment of the experts and patients shows the weighting of the symptom “nausea”. While the patients ranked this symptom in fourth place, the experts only placed this symptom on the seventh rank [14,15].

## 5. Discussion

### 5.1. Study Importance and Implications

In this study, 1096 affected patients were asked for their priorities and importance regarding different symptoms of functional dyspepsia and motility disorder. Based on the data analysis, the symptoms of abdominal cramps, vomiting and epigastric pain were most important and thus have the greatest influence on the well-being of the affected patients. In the middle range are the symptoms of nausea, acid reflux/indigestion, sickness, and retrosternal discomfort, whereas the symptoms causing the least impact on well-being are the feeling of fullness, early satiety and loss of appetite. The analysis and the comparison with the collected frequencies of occurrence showed that the symptoms that sometimes or often occur in most respondents have a significantly higher importance than the symptoms that rarely or never occur.

The results of the preliminary study within the expert survey yield different results to the patient survey. One deviation between the assessment of the experts and the patients is the weighting of the symptom “nausea”. This difference may reflect the potential relationship between the three symptoms of “nausea”, “sickness” and “vomiting”. In the expert panel, the participants stated that these symptoms cannot be clearly separated from each other and thus belong together. In contrast, it seems that here, the patients differentiate more clearly between these symptoms. One explanation might be that the patients have more experiences and empirical values regarding these gastrointestinal symptoms.

In general, the evaluation of the BWS has shown that patients weighted the ten gastrointestinal symptoms differently. Furthermore, the analysis of the frequencies of occurrence collected showed that the symptoms that sometimes or even often occur in most respondents in comparison have a significantly higher importance than the symptoms which rarely or never occur. Only the symptom of “early satiety” occurs in contrast relatively frequently but in the results, it is shown to be not so important for the patients. The weight of this symptom is very low because many subjects did not perceive “early satiety” as restrictive; in some cases, it was even viewed as positive. 

It may therefore be assumed that the backgrounds and experiences of respondents had a great influence on the decisions of the subjects. In many clinical trials, the change in the GIS over a defined period is the primary endpoint. [11,28] Measuring the impact of dyspeptic symptoms on patients’ daily lives in clinical trials requires validated preference-based instruments [9]. The evaluation of the BWS data showed that the common application of the GIS is limited in reflecting the reality of dyspeptic patients. Thus, the importance of individual symptoms could vary due to different backgrounds and experience. It has been found that the occurrence of the symptom “stomach cramps” has a greater negative impact on the well-being of the patient than the symptom “early satiety”.

In addition, in the experiment, changing information sets only revealed small differences between acute or chronic conditions during a treatment decision. Consequently, the information given about the kind of occurrence of the disease had seemingly no effect on the importance of different gastrointestinal symptoms or the influence on well-being. Regardless of whether patients suffer from acute or chronic complaints, the symptoms “stomach cramps” and “vomiting” have the strongest negative influence, and “early satiety” as well as “loss of appetite” have the lowest negative impact on the well-being of the patients.

The present study includes a very large number of patients enrolled. In total, 1096 patients affected with functional dyspepsia or gastrointestinal disorders were recruited for the stated preference survey. The sample size is a unique characteristic of the study. The number of subjects is even higher than the number of trapped patients in various clinical studies for the treatment of functional dyspepsia and gastrointestinal motility disorders [29,30,31,32,33].

As mentioned, the common GIS assumes that all 10 symptoms are weighted equally. This assumption was tested in this study. The results show that unlike the results of the GIS in the practice, the 10 symptoms of the GIS in this survey are not all weighted equally. So, the results show that the three symptoms of abdominal cramps, vomiting and epigastric pain are weighted considerably higher than symptoms such as the feeling of fullness, early satiety and loss of appetite.

It becomes clear that from a patient’s perspective, the common use of the GIS fails. In its current form, the GIS and its application in clinical studies or clinical practice does not represent a preference-based measurement instrument.

### 5.2. Limitations

The study has some limitations which must be considered in the evaluation, interpretation and generalization of the findings. The recruitment of respondents was accomplished by an external third-party provider. This could have influenced the included study population with respect to certain parameters, which could not be fully controlled. However, the access to market research companies was free for every patient and each person was able to enroll to the organization. In addition, preferences and priorities can be affected by various conditions. So, the information provided (features and specifications), the experience and background of the patient, as well as cognitive skills, are crucial [34,35]. Therefore, preferences and priorities may vary depending on the decision context. During the interpretation of the results, this must be taken into consideration. Preferences could even depend on the cultural background of the study samples and, for example, the existing health care system as a context factor [36]. Accordingly, when interpreting and generalizing the present study results, it should be considered that a German study sample was used.

## 6. Conclusions

Dyspeptic symptoms impair patients’ daily functioning [6,7,8]. The symptom-driven nature of patient management places particular importance on the reliable estimation of symptom status. Hence, the assessment of how symptoms of functional dyspepsia and motility disorder affect patients’ lives provides essential information about the patients’ health status and their perception of the treatment regime. Moreover, this information helps enable clinicians to tailor treatment to the individual patient’s needs [9].

Building on the ascertained weighting of the gastrointestinal symptoms and the evaluations from the patients’ viewpoint, further studies will be required to provide more details of patient preferences in this therapeutic area. The study design used in this survey can serve as a basis for more detailed preference studies for effectiveness research. The aim of this study was to provide a weighting, and therefore, a preference basing for an established measurement instrument for use in assessing the burden of disease in functional dyspepsia and motility disorders as part of a benefit–risk assessment. Our findings can offer an additional source of information and provide physicians and decision makers in health care with evidence on patients’ perspectives in the indication of gastrointestinal conditions [14].

To our knowledge, this is the first study which weights the symptoms of a standardized evaluation tool used in clinical studies and clinical practice regarding the assessment and measurement of the burden of illness in the field of GI conditions. Future studies must inform researchers and decision makers on how much a single gastrointestinal symptom impacts well-being from a patient’s perspective. This gap reflects the need to modify the GIS. The consideration of the results in clinical trials (with regard on the development of new drugs and treatments) as well as in the clinical practice can lead and improve patient-centered decisions in health care.

Future studies must calculate importance weights of patient-relevant symptoms to provide insights on the relative importance of individual symptoms. This will improve treatment decisions and provide information about the willingness of patients to weigh different symptoms. The findings of this study provide the first signs of a potential modification of the established GIS.

## Figures and Tables

**Figure 1 ijerph-18-11715-f001:**
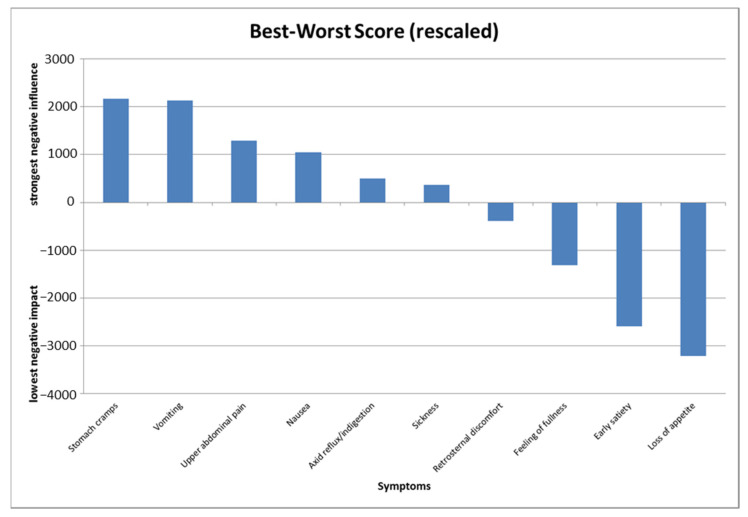
Best–worst scores (rescaled) of the 10 gastrointestinal symptoms.

**Figure 2 ijerph-18-11715-f002:**
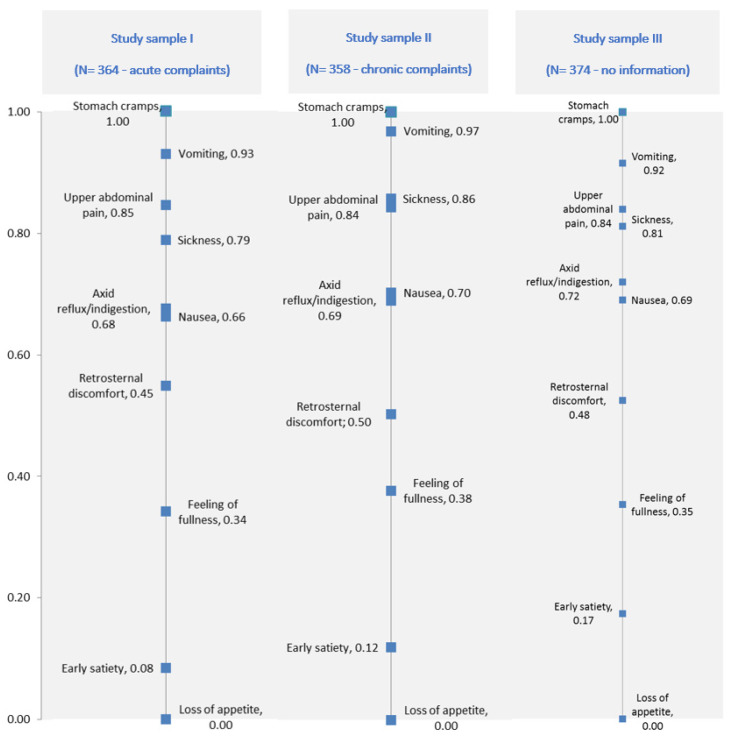
Weighting of gastrointestinal symptoms separated for each study sample.

**Table 1 ijerph-18-11715-t001:** Decision model for BWS case 1.

Symptom	Explanation
**Nausea**	Mood disorder that is also known as a “queasy” feeling in the stomach area and is accompanied by the urge to vomit.
**Vomiting**	Surge-like emptying of stomach contents through the mouth.
**Sickness**	Feeling of having to vomit and the immediate predecessor of vomiting.
**Feeling of fullness**	Bloated feeling, the supersaturation or overload of the stomach.
**Abdominal cramps**	Strong, colicky abdominal cramps, that decrease and increase repeatedly in their strength.
**Early Satiety**	Early onset feeling of overfilling the stomach. The saturation occurs immediately after ingestion.
**Acid Reflux/Indigestion**	From epigastric rising burning and painful sensation that can radiate to the neck and throat, often in connection with acidic or bitter regurgitation.
**Loss of Appetite**	Missing or limited need for food intake.
**Retrosternal Discomfort**	Unpleasant, painful or dragging sensation behind the breastbone.
**Upper Abdominal pain**	Pain that occurs between the costal arch, i.e., in the upper abdomen.

**Table 2 ijerph-18-11715-t002:** Socio-demographic characteristics of the study sample.

Characteristics	Absolute Number (%)
**Gender**	
Male	596 (54.4)
Female	500 (45.6)
**Age**	
20–29 years	109 (9.9)
30–39 years	264 (24.1)
40–49 years	322 (29.4)
50–59 years	266 (24.4)
>60 years	135 (12.4)
Mean/SD	45.3/11.7
**Material Status**	
Married	679 (62.0)
Widowed	26 (2.4)
Divorced or separated	103 (9.4)
Single	155 (14.1)
In a relationship, but not married	133 (12.1)
Others	0 (0.0)
**Employment Status**	
Employed full-time	818 (74.6)
Employed part-time	114 (10.4)
Self-employed	29 (2.6)
Homemaker	36 (3.3)
Student	32 (2.9)
Retired	51 (4.7)
Disabled/Unable to work	8 (0.7)
Unemployed but looking for work	7 (0.6)
Unemployed and not looking for work	1 (0.1)
**Body Heights (cm)**	
Mean/SD	174/75
**Weight (kg)**	
Mean/SD	77/24
**General State of Health**	
Very good	29 (2.6)
Good	443 (40.4)
Satisfactory	419 (38.2)
Not very good	168 (15.3)
Bad	37 (3.4)

**Table 3 ijerph-18-11715-t003:** Frequency of occurrence for the gastrointestinal 10 symptoms.

	Frequency of Occurrence	Never	Rarely (1 Time per Month or Less)	Occasionally (More Than 1 Time per Month)	Often (Several Times per Week)	Always (Every Day)
Symptom						
**Nausea**		227 (20.7%)	403 (38.8%)	354 (32.3%)	108 (9.9%)	4 (0.4%)
**Vomiting**		527 (48.1%)	355 (32.4%)	198 (18.1%)	16 (1.5%)	0 (0%)
**Sickness**		456 (41.6%)	369 (33.7%)	217 (19.8%)	54 (4.9%)	0 (0%)
**Feeling of Fullness**		170 (15.5%)	212 (19.3%)	382 (34.8%)	296 (27.0%)	36 (3.3%)
**Stomach Cramps**		147 (13.4%)	287 (26.2%)	399 (36.4%)	251 (22.9%)	12 (1.1%)
**Early Satiety**		211 (19.2%)	267 (24.4%)	424 (38.7%)	164 (15.0%)	30 (2.7%)
**Acid Reflux/Indigestion**		227 (20.7%)	224 (20.4%)	375 (34.2%)	242 (22.1%)	28 (2.6%)
**Loss of Appetite**		239 (21.8%)	379 (34.6%)	328 (29.9%)	137 (12.5%)	13 (1.2%)
**Retrosternal Discomfort**		361 (32.9%)	359 (32.8%)	289 (26.4%)	84 (7.7%)	3 (0.3%)
**Upper Abdominal Pain**		115 (10.5%)	272 (24.8%)	400 (36.5%)	283 (25.8%)	26 (2.4%)

**Table 4 ijerph-18-11715-t004:** Results of the BWS count analysis.

Attribute	Best Counts	Worst Counts	B–W Score	B–W Score Rescaled	Average B–W Score	Sqrt (B + 0.1/W + 0.1)	Sqrt (B + 0.1/W + 0.1) Rescaled
**Sickness**	537	897	−360	360	−0.08	−0.26	0.68
**Vomiting**	285	2418	−2133	2133	−0.49	−1.07	0.94
**Nausea**	356	1404	−1048	1048	−0.24	−0.69	0.82
**Feeling of Fullness**	1642	331	1311	−1311	0.30	0.80	0.36
**Stomach Cramps**	185	2359	−2174	2174	−0.50	−1.27	1.00
**Early Satiety**	2723	126	2597	−2597	0.59	1.54	0.13
**Acid Reflux/Indigestion**	636	1140	−504	504	−0.11	−0.29	0.70
**Loss of Appetite**	3279	66	3213	−3213	0.73	1.95	0.00
**Retrosternal Discomfort**	961	577	384	−384	0.09	0.26	0.53
**Upper Abdominal pain**	356	1642	−1286	1286	−0.29	−0.76	0.84

obs.: 21,920; *N* = 1096, r = 4.

## Data Availability

Data available on request.

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
