# Peer review of "The Impact of Gastrointestinal Symptoms on Patients’ Well-Being: Best–Worst Scaling (BWS) to Prioritize Symptoms of the Gastrointestinal Symptom Score (GIS)"

_ijerph, 2021, doi:10.3390/ijerph182111715_

Round 1

Reviewer 1 Report

  1. Are the Authors sure that the cited references are the most relevant and/or most recent? it would be better to update the references published in the nineties.
  2. Please review the definition of nausea. There is a difference between "mood disorder" and "feeling". Please justify the inclusion of these two different domains in the same definition. In addition, please consider that it semms not widely shared the inclusion of "urge to vomit" in the difinition of nausea. Again, justify.
  3. Pag.5, line 181 and following: it does not seem a good choice to combine the evaluation of the questionnaire regarding comprehensibility and feasibility and the survey itself. Please justify.
  4. Considering the very broad age range on included patients, it seems useful to connect marital staus (teenagers maybe are not widowed) and employment status (again teenagers are rarely retired) to age groups.
  5. The three different questionnaire versions were submitted randomly or to different kind of patients accordingly to their clinical status, that means those with acute complains received the version with the information acute complaints and so on? If the clinical status was not considered, please justify why this aspect was not relevant according to the Authors. This justification is even necessary considering the limitations presented by the Authors.

Author Response

Dear reviewers,

On behalf of my co-author, I would like to thank you and your fellow reviewers for your time and efforts in reviewing this manuscript. I am grateful for the quality reviewers and the support provided.

Based on these comments we have revised the whole manuscript.  While we have used track changes on additions and subtractions on the text, we did not do so for inserting references as this would have been unsightly.

Below you can find a point-by-point summary of how you have dealt with each of the comments raise during the peer review.

I trust that you will find the manuscript has improved and is now worthy of publication.

This said, if any further issues need addressing, we will quickly see that they are addressed, given that this is important and timely research.

With kind regards,

Axel Mühlbacher

Peer reviewer recommendations

Changes in the revised article

Reviewer 1

Are the Authors sure that the cited references are the most relevant and/or most recent? it would be better to update the references published in the nineties.

Thank you. During the process of finishing a manuscript one will always find new publications in the field. Hence, we had to decide which literature to use. Concerning the medical contexts, we decided to use some “standard literature”, which was also published in the nineties. The same is true for the metho of BWS. The underlying literature is rather old.

However, given your valuable comment, we deleted some redundancies in the references during the revision.

Please review the definition of nausea. There is a difference between "mood disorder" and "feeling". Please justify the inclusion of these two different domains in the same definition. In addition, please consider that it semms not widely shared the inclusion of "urge to vomit" in the difinition of nausea. Again, justify.

Thank you. Defining the attributes used in the experiment in a patient-friendly way is one of the most difficult things in the planning of the study. We wanted to make sure that all respondents have a common understanding of the attributes and levels. Hence, the definitions given in Table 1 are the “patient language”. For the definition of nausea we leaned on the definition of the National Cancer Institute (NCI Dictionaries). It states: “A feeling of sickness or discomfort in the stomach that may come with an urge to vomit. Nausea is a side effect of some types of cancer therapy.”

Pag.5, line 181 and following: it does not seem a good choice to combine the evaluation of the questionnaire regarding comprehensibility and feasibility and the survey itself. Please justify.

The final part of the survey included two questions on the ease of the questionnaire and the willingness to participate in another study of this format.

We agree that this might have been a bit misleading in the manuscript. We deleted this point.

Considering the very broad age range on included patients, it seems useful to connect marital staus (teenagers maybe are not widowed) and employment status (again teenagers are rarely retired) to age groups.

Thank you. This might be an interesting question for te discussion. Given the limited word count for the manuscript we cannot include this in the paper. Sorry.

The three different questionnaire versions were submitted randomly or to different kind of patients accordingly to their clinical status, that means those with acute complains received the version with the information acute complaints and so on? If the clinical status was not considered, please justify why this aspect was not relevant according to the Authors. This justification is even necessary considering the limitations presented by the Authors.

During the survey three different question versions were created with different BWS choice tasks and changing information sets on the severity of the symptoms. Participants received different information randomly. The different questionnaire versions were submitted randomly and not based on the clinical status. This would have answered another, also interesting research question. The aim of our study was to control for differing perceptions from a broader perspective.

The objective of this study was to prove that the underlying assumption of the GIS (which assumes that all ten gastrointestinal symptoms are weighted equally from a patients’ perspective) is wrong. The GIS does not consider any weighting of the included symptoms

To make this point clearer and to avoid any misunderstanding, we include a sentence in the paragraph 3.7. This reads as follows: The randomization was not linked to any personal information. This was used in refer-ence to the study hypothesis. In the GIS all ten gastrointestinal symptoms are weighted equally, independent on the clinical status of the patients. Using this survey approach this assumption should be tested using differing information sets, randomly assigned to patients with symptoms of varying severity.  .

Author Response

Dear reviewers,

On behalf of my co-author, I would like to thank you and your fellow reviewers for your time and efforts in reviewing this manuscript. I am grateful for the quality reviewers and the support provided.

Based on these comments we have revised the whole manuscript.  While we have used track changes on additions and subtractions on the text, we did not do so for inserting references as this would have been unsightly.

Below you can find a point-by-point summary of how you have dealt with each of the comments raise during the peer review.

I trust that you will find the manuscript has improved and is now worthy of publication.

This said, if any further issues need addressing, we will quickly see that they are addressed, given that this is important and timely research.

With kind regards,

Axel Mühlbacher

Peer reviewer recommendations

Changes in the revised article

Reviewer 2

A rather poor word order and overall wording

We revised the whole manuscript with special focus in wording and conciseness.

The reason of choosing the fractional factorial design over the full factorial design is not clear The explanation of using mentioned methodology is lacking in the article and adding it would be a requisite to improve legibility and conciseness of the presented paper.

A fractional factorial design was used to distribute the individual stimuli over the choice sets. This was used since a full factorial design would have place a too high cognitive burden on the respondents. Given the high number of attributes and levels a full factorial design would have resulted in too many choice sets.

However, the only pre-research phase mentioned in the article was carried out on 20 experts from the gastroenterological field, which is certainly not high enough sample size to utilize the method designed to control the redundancy of the full factorial design. If it is so, authors risk a data-validity loss with picking only a chosen subset.

Correct, the pre-research phase was carried out with 20 experts. This phase included tests of comprehensibility, manageability, and all-inclusive information. The pre-research phase did not measure any preferences.

For the final BWS, the chosen fractional factorial design is a given standard in choice experiments. It is true that there might be a certain risk of a data-validity loss. However, this is manageable by including a higher number of respondents in the final sample. Given this fact, we decided to include more than 1000 patients. This minimizes to risk of validity loss.

Paper’s sections related to the experimental design are to be explained more thoroughly

Thank you. We added additional information on the experimental design.

On various occasions the paper appears lengthy and forcefully extensive, with basically the same descriptive data stretched out to 3 tables and 2 figures

Thank you. During the revision we deleted lengthy paragraphs. Moreover, we restructured several sections to make it clearer, especially the experimental design section.

The clinical relevance of such scoring is missing. The hypothesis how the observed results could be used consequently in the clinical practice should be added. 

To our knowledge, this is the first study which weights the symptoms of a standardized evaluation tool used in clinical studies and clinical practice regarding the assessment and measurement of the burden of illness in the field of GI conditions. The study design used in this survey can serve as a basis for more detailed preference studies for effectiveness research. Our findings can offer an additional source of information and provide physicians and decision makers in health care with evidence on patient’s perspective in the indication of gastrointestinal conditions